# Long Exposure Short Pulse Synchronous Phase Lock Method for Capturing High Dynamic Surface Shape

**DOI:** 10.3390/s20092550

**Published:** 2020-04-30

**Authors:** Weiqiang Han, Xiaodong Gao, Zhenjie Fan, Le Bai, Bo Liu

**Affiliations:** 1Institute of Optics and Electronics of Chinese Academy of Sciences, Chengdu 610209, China; 13980523631@139.com (X.G.); fanzhenjie1984@163.com (Z.F.); BL511690069@163.com (L.B.); boliu@ioe.ac.cn (B.L.); 2Key Laboratory of Science and Technology on Space Optoelectronic Precision Measurement, CAS, Chengdu 610209, China; 3University of Chinese Academy of Sciences, Beijing 100149, China

**Keywords:** vibrating mirrors, pulsed laser, laser repeat frequency, synchronous phase lock, short pulse, interference pattern

## Abstract

In infrared weak target detection systems, high-frequency vibrating mirrors (VMs) are a core component. The dynamic surface shape of the VM has a direct impact on imaging quality and the optical modulation effect, so its measurement is necessary but also very difficult. Measurement of the dynamic surface shape of VMs requires a transiently acquired image series, but traditional methods cannot perform this task, as, when the VM is vibrating at a frequency of 3033 Hz, using high-speed cameras to acquire the images would result in frame rates exceeding 1.34 MFPS, which is currently technically impossible. In this paper, we propose the long exposure short pulse synchronous phase lock (LSPL) method, which can capture the dynamic surface shape using a camera working at 10 FPS. In addition, our proposed approach uses a single laser pulse and can achieve the dynamic surface shape measurement on a single frame image.

## 1. Introduction

High-frequency vibrating mirrors (VMs) are commonly utilized to suppress background light during small infrared target detection. However, the dynamic surface shape of the VM may directly affect the imaging quality and optical modulation effect of the optical system. Therefore, the accurate measurement of the surface of the vibrating mirror is important. Generally speaking, three-dimensional (3D) surface measurements are divided into two main categories [1,2,3,4,5,6]: contact and non-contact methods. The main advantage of contact methods is that they can achieve micron-level accuracy. Their main disadvantages are that they damage the target’s surface easily, they cannot be used to obtain subtle features of complex target surfaces, they are slow, and it is easy to wear the probe, which will degrade the measurement accuracy and shorten its service life. Non-contact methods do not require contact with the target, thereby avoiding surface damage.

Among the various non-contact 3D surface measurement methods, optical methods [7,8,9,10,11,12,13] are widely used in object surface measurement. In general, optical non-contact methods can be classified into laser ranging, structured light, and interference methods. Specifically, laser ranging methods obtain an object’s 3D image through direct or indirect measurement of the travel time and work with a scanning mechanism. The structured light methods project a prepared pattern image on the object’s surface, and the surface contour of the object modulates it. Then, a 3D image of the object is reconstructed from the deformed reflection image. However, the speed of laser ranging methods and structured light methods is so low that they are not suitable for dynamic measurements. Compared with laser ranging and structured light methods, interference methods [14,15] provide a possible way to take dynamic measurements due to their advantages of real-time performance and high accuracy.

There are three classic ways to implement interference methods. These include the fringe scanning method [16], the three interferogram method [17], and the single interferogram method [18]. In the fringe scanning method, the reference mirror is shifted so that the light intensity at any point in the interferogram is modulated by a sine function, and then the surface to be measured is derived using the Fourier transform. In the three interferogram method, three interferograms with different reference phase positions are acquired, corresponding to three kinds of intensity distributions with different phases of the reference light. In fact, both the fringe scanning method and the three interferogram method require a scanning mechanism to obtain sequence images. As a result, it is infeasible to apply them to fast-moving VM surface measurements due to the dynamic nature of the measurements.

In the single interferogram method, the object’s surface shape can be obtained from static interference patterns, but the accuracy is lower than the fringe scanning method and the three interferogram method. Furthermore, the single interferogram method requires the user to obtain the interference patterns with a high-frame-rate camera. Unfortunately, a VM with a frequency of 3033 Hz requires a camera frame rate of 1.34 MFPS to acquire the interference patterns, which is almost impossible to implement.

Some 3D imaging measurement methods use scattered light on the object’s surface for imaging. Related studies include short-time gate suppression scattered light imaging [19,20,21,22,23], wavefront imaging [24,25], speckle correlation imaging [26,27,28], and ghost imaging [29,30,31,32]. The imaging frequency of such methods is relatively low. Zi-Hao Xu’s method increases the frequency to 1000 fps [33], but the corresponding resolution is only 32 × 32 pixels, and spatial light modulation is required. All these imaging measurement methods are used on rough surfaces that can scatter light, and are not applicable for smooth mirror surfaces such as those of a VM.

Chen et al. used electronic speckle pattern interferometry (ESPI) to measure four vibration modes of a corrugated plate excited by a harmonic wave of a 550 Hz speaker [34]. De Veuster et al. used digital speckle pattern interferometry (DSPI) [35] to measure the amplitude of a diaphragm driven by a speaker at a vibration frequency of 1000 Hz. This type of speckle interference method is mainly aimed at rough surfaces and is suitable for interferometric imaging measurements of vibration nodes in vibration modes, rather than surface shape measurement.

There are also methods that deploy 3D camera technology, such as stereo vision [36,37]. Stefan Heist combined the structured light method and stereo vision technology to develop a high-speed projection imaging technology [38]. Using two cameras with a frame rate of up to 12,500 fps, an image of the airbag explosion process with a frame rate of 1333 Hz was obtained. This method requires expensive equipment, and the measurement accuracy is suitable for the measurement of dynamic processes, but the accuracy is too low to allow for the measurement of the VM surface shape. The method proposed in this paper can collect the surface interference pattern of a smooth mirror in vibration. Compared with Stefan Heist’s method, the method in this paper can use low-cost commercial cameras to obtain higher accuracy VM interference patterns at a frame rate as low as 10 fps.

In this paper, we propose a long exposure short pulse synchronous phase lock (LSPL) method, which allows the user to capture the surface interferogram of the VM instantaneously. The LSPL method uses the phase shifter to shift the position signal of the VM to the corresponding sampling position to be measured, and this generates the position synchronization signal (PSS), which synchronizes the laser pulse and camera. Through the strict synchronization between the sampling position of the VM, the laser pulse, and the camera, the camera can acquire the interference pattern generated by the laser pulse on the VM.

The rest of this article is arranged as follows. In the second section, the method of LSPL is described in detail. In the third section, the design and construction of the experimental setup are described. In Section 4, the experimental procedure and experimental results are presented.

The list of nomenclature:
LSPL: Long Exposure Short Pulse Synchronous Phase LockVM: vibrating mirrorLPW: laser pulse widthSL: sampling locationIOP: Interference optical path

## 2. Approach Overview

The method is called long exposure short pulse synchronous phase lock (LSPL). Figure 1a depicts the signal transmission relationship. The one-dimensional turntable rotates at the corresponding angle according to the sampling location (SL) of the VM to be measured. At the same time, the phase shifter shifts the corresponding phase according to SL(i) to generate the position synchronization signal (PSS) for the laser and camera as a synchronization signal, where “*i”* is an index denoting one of the total number (*Tn*) of sampling locations. After receiving the PSS, the laser emits the pulse towards the VM via the interference optical path (IOP) to generate an interference pattern at the VM nearby. The pulse carrying the interference pattern enters the focal plane of the camera after passing a part of the IOP, and the interference pattern is captured by the camera. The camera uses a long exposure time at the SL(i) position for the acquisition, and the laser pulse width (LPW) is short enough to produce the interference pattern, as shown in Figure 1b.

The dwell time (*dt*) is a very important factor for the LSPL method; to know what factors affect it, we must analyze the VM’s motion characteristic as follows.

The tested VM is shown in Figure 2a. The vibration mode of the VM is a sinusoidal swing around the axis as shown in Figure 2b, the frequency of VM vibration is defined as $f_{n}$, the amplitude as $B_{n}$, and the initial phase as $\varphi_{2}$. The angular position (θ) of the vibration is expressed using Equation (1).
(1)$$\theta =B_{n}\sin\left(2\pi f_{n}t+\varphi_{2}\right)$$

Consequently, the angular velocity (v) is:(2)$$v=2\pi f_{n}B_{n}\cos\left(2\pi f_{n}t+\varphi_{2}\right).$$

And the angular acceleration (a) is:(3)$$a=-4\pi^{2}f_{n}^{2}B_{n}\sin\left(2\pi f_{n}t+\varphi_{2}\right).$$

The relevant parameters of the VM are shown in Table 1.

Depending on the VM’s angular position as given in Equation (1), the *SL*(*i*) position within a period varies between −*B_n_* and *B_n_*, and there are infinite positions in-between, so the selection of the total number of the *SL*(*i*) is important for the implementation of the LSPL. The total number (*Tn*) selection principle is that the captured interference pattern is not blurred. The criterion is the blur angle (dθ) of the interference patterns, which is expressed by the following equation:(4)$$d\theta =\tan^{-1}\left(\frac{n\lambda }{2D_{r}}\right)$$
where $D_{r}$ is the interference beam diameter, n is the number of interference patterns, and λ is the wavelength of the laser pulse. From Equation (1), we see that the blur angle (dθ) and the dwell time (dt) satisfy the following relationship:(5)$$d\theta =B_{n}\sin\left[2\pi f_{n}\left(t+dt\right)+\varphi_{2}\right]-B_{n}\sin\left(2\pi f_{n}t+\varphi_{2}\right).$$

For simplicity, let the initial phase ($\varphi_{2}$) be 0. Then:
(6)$$\frac{d\theta }{B_{n}}=\sin\left[2\pi f_{n}\left(t+dt\right)\right]-\sin\left(2\pi f_{n}t\right).$$

This can be decomposed into:
(7)$$\frac{d\theta }{B_{n}}=\sin\left(2\pi f_{n}t\right)\cos\left(2\pi f_{n}dt\right)+\cos\left(2\pi f_{n}t\right)\sin\left(2\pi f_{n}dt\right)-\sin\left(2\pi f_{n}t\right).$$

When dt is small enough, the equation can be simplified to:
(8)$$\frac{d\theta }{B_{n}}=\sin\left(2\pi f_{n}t\right)+\cos\left(2\pi f_{n}t\right)\sin\left(2\pi f_{n}dt\right)-\sin\left(2\pi f_{n}t\right)$$
(9)$$\frac{d\theta }{B_{n}}\approx \cos\left(2\pi f_{n}t\right)\sin\left(2\pi f_{n}dt\right)\approx 2\pi f_{n}dt\cos\left(2\pi f_{n}t\right)$$
and further to:
(10)$$d\theta \approx \left[2\pi B_{n}f_{n}\cos\left(2\pi f_{n}t\right)\right]dt.$$

We combine it with the angular velocity definition of Equation (2) to obtain the following equation:
(11)$$dt\approx \frac{d\theta }{v}.$$

The above simplified equation is suitable for small *dt*. Figure 3a shows the relationship between the dwell time and the blur angle, where it can be seen that the dwell time is long at the position far from the center because of the low speed and short at the center because of the large speed. To obtain the interference pattern within the blur angle during the full cycle of the VM, the capturing time of the camera needs to be less than *dt,* which is not achievable by commercial-grade imaging circuits. In this article, we show how the LSPL method utilizes a short pulse (LPW less than *dt*) laser to generate the interference pattern and a camera with a long exposure time to capture it.

During the exposure time of the camera, the interferogram of the pulsed laser light reaches the focal plane of the camera and is acquired by the camera. This enables us to capture the interference pattern in a short time and obtain the interference pattern at this *SL*(*i*). A one-dimensional table is used to compensate for the offset during VM vibration. The one-dimensional table rotates at angle *θ*, and the corresponding *SL*(*i*) is at angle *θ*. The total number (Tn) of sampling locations (SLs) is
(12)$$Tn=\frac{2B_{n}}{d\theta }.$$

To ensure that the interference pattern is accurately acquired within the dwell time at the corresponding *SL*(*i*), the synchronization accuracy ($t_{P}$) of the PSS is very critical. The synchronization relationship is shown in Figure 3b. Because *dt* and LPW are very small compared with the exposure time, to provide a better explanation of the synchronization relationship, the space ratio in Figure 3b is bigger than Figure 1b. When the VM swings to the *SL*, the laser pulse must be able to reach the VM. It will take a certain amount of time for the laser pulse from the laser to reach the VM, and it will take a certain amount of time to transmit from the position where the interference occurs to the camera. Therefore, in order to acquire the dynamic surface interference pattern of the VM, when the VM vibrates to the *SL*(*i*), the pulse emitted by the laser will pass through the IOP to reach the VM, and the two need to be strictly synchronized. Interference occurs near the VM; when the pulse with the interference pattern returns, it passes through the subsequent IOP and enters the focal position of the camera, where it is recorded within the exposure time.

To be able to obtain interference images on the camera, the pulse must reach the VM within *dt* for each *SL*. Therefore, as shown in Figure 3b, the overlap time (*OT*) of the LPW and *dt* should be greater than the time taken by the light pulse to travel through the IOP. This is expressed by the following equation:(13)$$OT=\left(t_{SL}+dt\right)-\left(t_{SL}+t_{s}+t_{P}+t_{w}\right)>\frac{c}{OP}.$$

In the above equation, $t_{SL}$ is the moment when the *SL* has just arrived, $t_{S}$ is the time delay related to SL(i) caused by the phase shifter, and $t_{P}$ is the duration of the rising edge of the PSS pulse. The time delay ($t_{w}$) is caused by the length of the wire between the phase shifter and the laser. $\frac{c}{OP}$ is the time taken by the laser pulse from the laser to reach the focal plane of the camera and cross the optical path, where *c* is the speed of light, and *OP* is the length of the IOP. The *SL* is an integer multiple of the blur angle and is expressed as follows:(14)SL(i)=idθ,i=−Tn/2⋯0⋯Tn/2.

So, the time delay ($t_{s}$) corresponds to the SL(i) and is expressed as follows:(15)$$t_{s}\left(i\right)=iLI,i=-Tn/2\cdots 0\cdots Tn/2.$$

The location interval (*LI*) is the time interval between each *SL*, which is described as: (16)$$LI=\frac{1}{f_{n}T_{n}}.$$

Since the time delay ($t_{s}$) is an integer multiple of the *LI*, its influence on the synchronization relationship can be ignored. From Equation (13), the synchronization accuracy ($t_{P}$) of the PSS can be obtained as shown in Equation (17).
(17)$$t_{P}<dt-\frac{c}{OP}-t_{w}$$

The constraint conditions to capture the dynamic surface shape using the LSPL approach are as follows.
The laser pulse must arrive at the VM just at the moment that the VM is reaching the SL(i).The interference pattern must be generated at the blur angle (dθ) within the dwell time (dt). The laser pulse controlled by the PSS must be strictly aligned with the *SL*, so the accuracy of the PSS must meet the condition that the synchronization accuracy is $t_{P}<dt-\frac{c}{OP}-t_{w}$.Only one laser pulse is generated within the period of the camera’s exposure time to produce one interference pattern, otherwise the interference fringe captured by the camera will be blurred.

## 3. Experimental Setup

Two experimental systems were set up for comparison, one with a measurement system built according to the method proposed in this paper in Figure 4a, and a second built using a traditional interferometer, as shown in Figure 4d.

Figure 4a shows the experimental system built for applying the LSPL method proposed in this paper, and the corresponding parameters are shown in Table 1, where the laser is a product of the company THORLABS. The IOP is shown in Figure 4b. The computer and controller are used to control the one-dimensional turntable and camera shown in Figure 4c.

The frequency of the VM was $f_{n}=3033\;\mathrm{Hz}$, with an amplitude $B_{n}=2.6\times 10^{-3}\;\mathrm{rad}$. From Equations (2) and (3), we calculated the maximum angular velocity (MAV) as $48.8\;\mathrm{rad}\cdot s^{-1}$, and the max angular acceleration (MAA) as $9.3\times 10^{5}\;\mathrm{rad}\cdot s^{-2}$. The laser wavelength was 532 nm, the interference aperture was 56 mm, and the number of interference patterns was 5, so the blur angle dθ was calculated as $2.3\times 10^{-5}\;\mathrm{rad}$ from Equations (4) and (11). The minimum dwell time dt was 476 ns. *Tn*, as calculated using Equation (12), was 221 and the location interval, calculated using Equation (16), was 1494 ns.

The LPW was 129 ns and the pulse’s frequency was controlled using the PSS. The transmission distance of the PSS signal was limited to within 3 m, so $t_{w}<9.9\;\mathrm{ns}$. The propagation path (OP) was not greater than 2 m, so $\frac{c}{OP}<6.6\;\mathrm{ns}$. Therefore, the synchronization accuracy ($t_{P}$) of the PSS was calculated from Equation (17) as 459.5 ns. The accuracy of the phase shifter used was 90 ns, which meets the accuracy requirement of less than 495.5 ns.

Within an exposure time of 100 ms, the VM vibrates for 303 cycles, and each cycle has 221 positions, which is equivalent to accurately locating one of the 66,963 continuously changing positions.

## 4. Results

This section describes the experiments that were conducted to verify the method. First, we present the experiments with the traditional interferometer approach, and we then follow with the experimental results using the LSPL approach.

### 4.1. Experimental Results with the Traditional Interferometer

VM1 was installed on fixed tooling. Because it was intended only for comparative experiments, the surface shape of VM1 was not adjusted to its optimal shape. The interferometer was mounted on a two-dimensional tilt adjustment mechanism to facilitate the adjustment of the pitch and horizontal angles. With VM1 at a standstill, we adjusted the two-dimensional tilt adjustment mechanism so that interference patterns were observable on the screen. The fringes can be clearly seen in Figure 5a. When the vibration of VM1 was started, the fringes disappeared and could not be measured, as shown in Figure 5b. Therefore, traditional interferometry methods cannot perform dynamic surface shape measurements on VMs.

### 4.2. Experimental Results Using LSPL

With the VM at a standstill, the one-dimensional turntable was set at the zero position and the LPW was set to 129 ns. The interference patterns captured are shown in Figure 6a. Three interference patterns—S1, S2, and S3—were obtained in one frame of the image. Fixed pattern FG1 and FG2 in the figure are fixed patterns introduced by the surface reflection of the polarizing beam splitter (PBS). Although S3 is not very clear, the three interference patterns—S1, S2, and S3—can be individually adjusted for contrast. We started the VM’s vibration with the one-dimensional turntable at the zero position, corresponding to the sampling position *SL*(0). We then adjusted the PSS frequency to 10 Hz, 20 Hz, 30 Hz, 50 Hz, and 100 Hz, and we acquired corresponding interference pattern images, as shown in Figure 6. The patterns in Figure 6b are clear, but the patterns in Figure 6c–f are blurred. The camera’s acquisition frame rate was 10 FPS, and the PSS = 20 Hz, which means that the laser emits two pulses during the exposure time of the camera to generate two interference patterns. Each frame image is the energy superposition of interference patterns generated by the two laser pulses. Because the width of a single laser pulse is 129 ns, two such pulses are separated by 50 ms, and each pulse is separated by a considerable distance, so the fringe blur in Figure 6c is not caused by pulse crosstalk but by VM vibration. The blurring of Figure 6d–f is due to the same reason. Therefore, the single frame and single laser pulse method must be used for measurement of the surface shape during VM vibration, as in Figure 6b. This also verifies the feasibility of the LSPL method proposed in this paper.

Figure 6 shows the interference pattern obtained by sampling at the *SL(*0) position of the VM. We then sampled at both ends, i.e., *SL*(−110) and *SL*(110). The experimental results, shown in Figure 7, show that the LSPL method can be used to realize dynamic surface shape measurement during VM vibration cycles.

## 5. Discussion and Conclusions

In this paper, we proposed an approach to capturing the dynamic surface shape of a VM using a camera working at a 10 Hz frame rate. We draw the following conclusions:Unlike high-speed projection-imaging technology that uses expensive high-frequency cameras to capture dynamic images, our proposed method uses inexpensive commercial cameras to acquire the surface interference pattern vibrating at 3033 Hz using a frame rate as low as 10 fps.Acquiring the interference pattern at the sample location *SL*(*i*) has two meanings: (1) the one-dimensional turntable rotates at the corresponding angle according to the sampling location (*SL*) of the VM to be measured; and (2) the phase shifter moves a phase offset corresponding to *SL*(*i*) and a PSS is generated. The PSS synchronizes the laser pulse and the camera for the generation and acquisition of the interference fringe of the VM at *SL*(*i*). Since the frame frequency of the camera is 10 Hz, each image frame captured by the camera is separated by 303.3 VM vibration cycles, but they are all interference patterns captured at a fixed position *SL*(*i*) of each vibration cycle. When the turntable rotates from *SL*(−110) to *SL*(110) (a total of 221 *SL*s), all positions within a single vibration cycle of the VM are measured.Compared to the traditional interference methods, the proposed approach can realize dynamic surface shape measurement while the VM is vibrating, while the traditional interference method failed.In addition, the difference between our proposed approach and the traditional interference method is that the laser we used is a pulsed laser that can capture the dynamic surface shape on a single frame image using a single pulse.The proposed approach adopts the method of LSPL, where a PSS forms a nanosecond-precision phase-lock relationship between the VM’s phase and the laser pulse and camera acquisition.The constraint conditions for capturing the dynamic surface shape must be satisfied, such as “the laser pulse controlled by the PSS must be strictly aligned with the SL”.The experiment shows that the VM’s dynamic surface shape can be captured using only a single pulse within the period of the camera’s exposure time. If more than one pulse is emitted within the period of the camera’s exposure time, the interference patterns will be blurred.The experimental results verify the correctness of the LSPL approach.

## Figures and Tables

**Figure 1 sensors-20-02550-f001:**
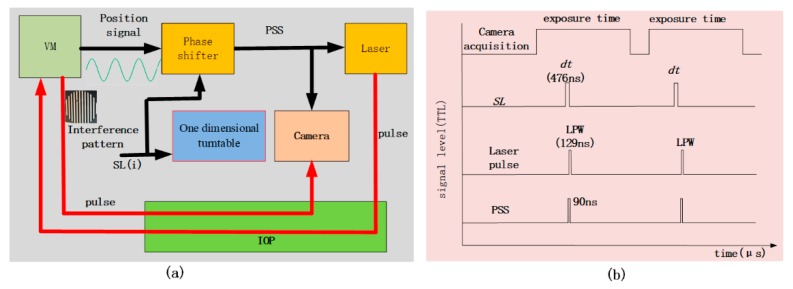
(**a**) Signal flow. The red line represents optical paths, and the black line represents electronic paths. (**b**) Synchronization relationship.

**Figure 2 sensors-20-02550-f002:**
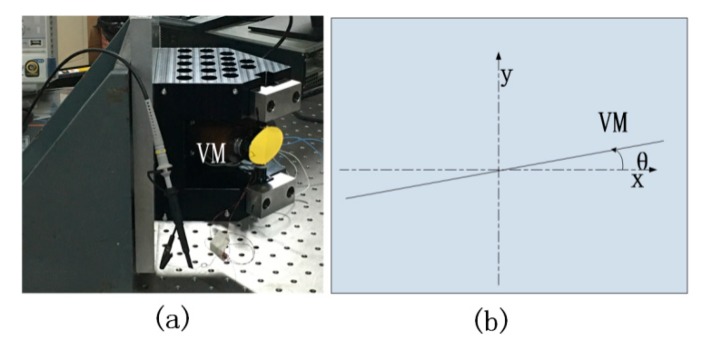
(**a**) Vibrating mirror (VM) diagram. (**b**) Coordinate system.

**Figure 3 sensors-20-02550-f003:**
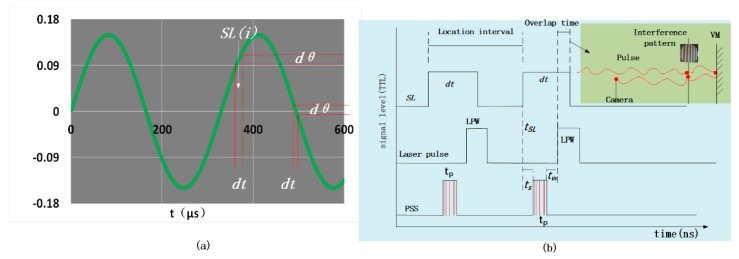
(**a**) Relationship between dwell time (*dt*) and blur angle (dθ). (**b**) The synchronization relationship.

**Figure 4 sensors-20-02550-f004:**
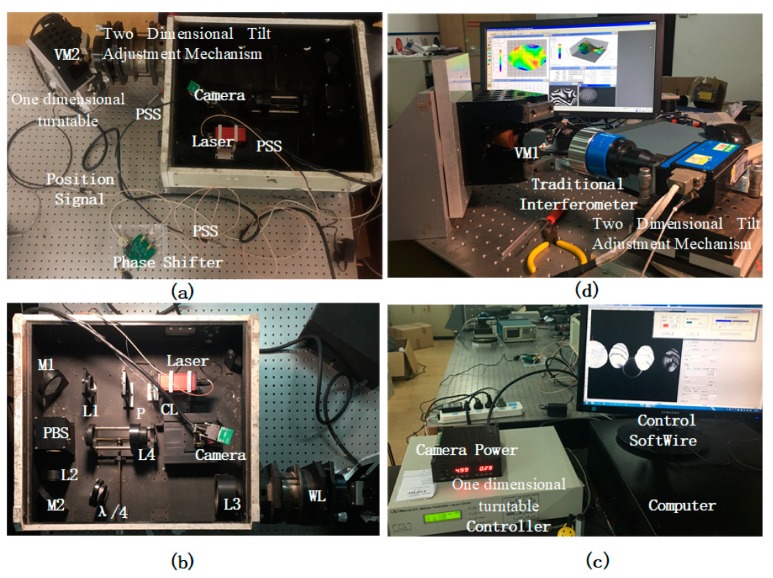
Experimental setup. (**a**) Experiment setup using long exposure short pulse synchronous phase lock (LSPL). (**b**) Interference optical path (IOP) setup in the laboratory. (**c**) Computer and controller. (**d**) The experiment setup using a traditional interferometer.

**Figure 5 sensors-20-02550-f005:**
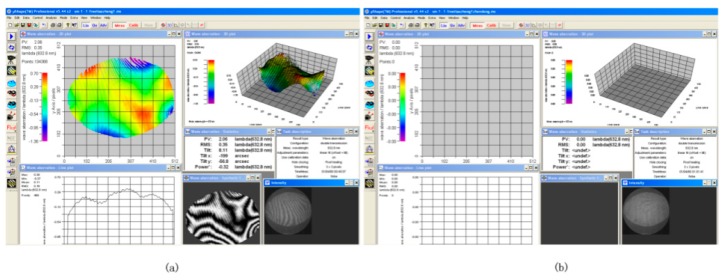
Experimental results using the traditional interferometer approach. (**a**) Interference pattern on a static VM. (**b**) No interference pattern on a vibrating VM.

**Figure 6 sensors-20-02550-f006:**
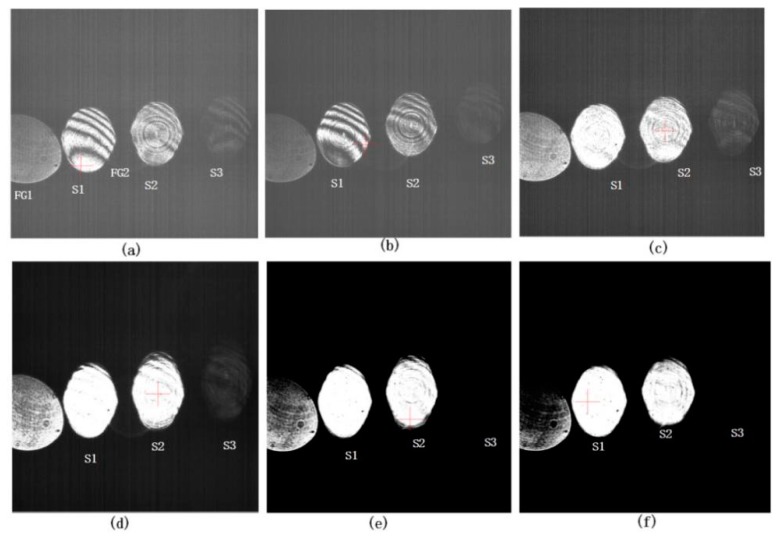
Experimental results using the LSPL approach at *SL*(0). (**a**) Interference pattern with a static VM. S1, S2, and S3 are the interference patterns on a single image. (**b**) Interference pattern with a vibrating VM at position synchronization signal (PSS) = 10 Hz. (**c**) Interference pattern at PSS = 20 Hz. (**d**) Interference pattern at PSS = 30 Hz. (**e**) Interference pattern at PSS = 50 Hz. (**f**) Interference pattern at PSS = 100 Hz.

**Figure 7 sensors-20-02550-f007:**
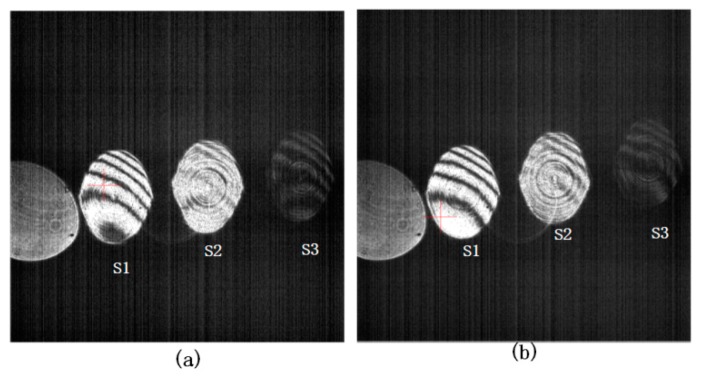
Experimental results using the LSPL approach. (**a**) Interference pattern with the VM in a vibrating state when PSS = 10 Hz, *SL*(−110). (**b**) Interference pattern when PSS = 10 Hz, *SL*(110).

**Table 1 sensors-20-02550-t001:** Parameters of the Long Exposure Short Pulse Synchronous Phase Lock (LSPL) experiment setup.

Parameter	Value	Unit
VM Frequency $f_{n}$	3033	Hz
VM Amplitude $B_{n}$	$2.6\times 10^{-3}$	rad
MAV	48.8	$\mathrm{rad}\cdot s^{-1}$
MAA	$9.3\times 10^{5}$	$\mathrm{rad}\cdot s^{-2}$
exposure time	100	ms
Camera Frame Rate	10	fps
LPW	129	ns
Laser wavelength λ	532	nm
dθ	$2.3\times 10^{-5}$	rad
dt	476	ns
Location interval	1494	ns
OP	2	m
$t_{P}$	90 (<495.5)	ns
Tn	221

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
