# Peer review of "Long Exposure Short Pulse Synchronous Phase Lock Method for Capturing High Dynamic Surface Shape"

_sensors, 2020, doi:10.3390/s20092550_

Round 1

Reviewer 1 Report

The manuscript under the title “Long-exposure short-pulse synchronous-phase-lock method for capturing high dynamic surface shape” describes the implementation of the Long Exposure Short Pulse Synchronous Phase Lock (LSPL) method that has a capability to capture the dynamic surface shape via a camera. The experimental analyses were carried out that proved the trustworthiness of the LSPL method. The study is really interesting and unique, as well. However, I found the following issues regarding the manuscript:

  1. There is no need to use the abbreviation of the “Long Exposure Short Pulse Synchronous Phase Lock method” in the Abstract section as it is not going to be used furthermore in the Abstract section.
  2. Please revise the second last sentence in the Abstract section as it is not making any sense.
  3. It will be better to provide a list of nomenclature (abbreviations used in the whole manuscript) after the introduction section. This can avoid the long captions under the figures (Figs. 1, 3, and 4).
  4. There is a strong need to redefine the introduction part as I didn’t find any previous studies and what’s the difference between your work and the previous studies. The current introduction has solely described measuring methods and ways to implement interference methods.
  5. Please revise the sentence “The dwell time (dt) is a very import factor to the LSPL method; to determine it, we must analyze the VM’s motion characteristic as follows.”, as the sense of this sentence is not clear.
  6. Please present the conclusion in bullets form as it will assist the audience in understanding the outcome of the study in a better way.

Reviewer 2 Report

Review for "Long-exposure short-pulse synchronous-phase-lock method
for capturing high dynamic surface shape" by Han, Gao, Fan, Bai, and Liu

This paper shows how to detect the vibrations of a vibrating mirror (VM) using laser interferometry, and a camera with slow frame capture rate. A narrow laser pulse and precision timing allows for the fast VM vibrating at around 3kHz to be captured by a imaging camera that captures 10 frames per second. The authors give guidelines about the timing of the laser pulse and the camera imaging. 

While the results are interesting  I do think that the manuscript would need to be improved substantially before publishing.

  • The Introduction needs more details about the new technique, just referring to by name is not sufficient.
  •  
  • The authors over-rely on acronyms in Section 2. The authors should limit themselves to five acronyms. Most are only used two or three times. VM and the variables in the equations are fine. The over-use of other acronyms  makes the paper to be illegible. 
  •  
  • Figure 1 needs improvement. there are many different optical and electronic paths in this figure. The authors should distinguish between optical and electronic paths. Also there are pulses created that should be explicitly shown in this figure.
  •  
  • There is a discrepancy between figure 1(b) and figure 3(b). the mark to space ratio is bigger in figure 3(b) and figure1(a) , this may be done for illustration purposes though should be explained clearly.

In figure 6(a), does the camera capture the three interference patterns simultaneously? Or is the figure a composite of different captures?

To the best of my reading, the authors show a single frame capturing one interference pattern; have the authors got their system to work to capture  the full sequence across a single cycle of the vibrating mirror?

Can the authors state in the paper about the laser pulse source, is this a commericially available source? is the laser directly modulated or do they use an external modulator to create the laser source.

Round 2

Reviewer 1 Report

Thank you for considering our suggestions. We recommend publishing the article in its current form.

Author Response

Point 1:Thank you for considering our suggestions. We recommend publishing the article in its current form.

Response 1:Thank you for your patient comments and suggestions.

Reviewer 2 Report

The revisions are fine, I recommend the paper be published as is.

Author Response

Point 1: The revisions are fine, I recommend the paper be published as is.

Response 1:Thank you for your patient comments and suggestions.